Estimating the average daily rainfall in Thailand using confidence intervals for the common mean of several delta-lognormal distributions

Maneerat Patcharee 1
Niwitpong Sa-Aat sa-aat.n@sci.kmutnb.ac.th 2
1 Department of Mathematics, Faculty of Science and Technology, Uttaradit Rajabhat University , Uttaradit , Thailand
2 Department of Applied Statistics, Faculty of Applied Science, King Mongkut’s University of Technology North Bangkok , Bangkok , Thailand
Raga Graciela
Electronic publication date: 2021 Jan 22
Publication date: 2021
Volume: 9
Electronic Location ID: e10758
Received 2020 Aug 11; Accepted 2020 Dec 21
Copyright: ©2021 Maneerat and Niwitpong
Copyright year: 2021
Copyright holder: Maneerat and Niwitpong
License: This is an open access article distributed under the terms of the Creative Commons Attribution License, which permits unrestricted use, distribution, reproduction and adaptation in any medium and for any purpose provided that it is properly attributed. For attribution, the original author(s), title, publication source (PeerJ) and either DOI or URL of the article must be cited.
License URL: https://creativecommons.org/licenses/by/4.0/

Keywords: Agriculture, Bayesian approach, MOVER, Natural rainfall, Vague prior, Variance

Funding: King Mongkut’s University of Technology North Bangkok KMUTNB-BasicR-64-26 This research was funded by King Mongkut’s University of Technology North Bangkok (grant number: KMUTNB-BasicR-64-26). The funders had no role in study design, data collection and analysis, decision to publish, or preparation of the manuscript.

==============================
The daily average natural rainfall amounts in the five regions of Thailand can be estimated using the confidence intervals for the common mean of several delta-lognormal distributions based on the fiducial generalized confidence interval (FGCI), large sample (LS), method of variance estimates recovery (MOVER), parametric bootstrap (PB), and highest posterior density intervals based on Jeffreys’ rule (HPD-JR) and normal-gamma-beta (HPD-NGB) priors. Monte Carlo simulation was conducted to assess the performance in terms of the coverage probability and average length of the proposed methods. The numerical results indicate that MOVER and PB provided better performances than the other methods in a variety of situations, even when the sample case was large. The efficacies of the proposed methods were illustrated by applying them to real rainfall datasets from the five regions of Thailand.

Introduction

Approximately 82.2% of Thailand’s cultivated land area depends on natural rainfall (Supasod, 2006), thereby indicating its importance for Thai agriculture. However, it is a natural phenomenon with a significant level of uncertainty that can cause natural disasters such as droughts, floods, and landslides. In many countries around the world, extreme rainfall events have been increasing in frequency and duration. On December 5, 2017, Storm Desmond led to heavy rainfall causing flooding in northern England, Southern Scotland, and Ireland (Otto & Oldenborgh, 2017). On July 6–7, 2018, extreme rainfall events such as floods and landslides affected over 5,000 houses, and approximately 1.9 million people in Japan were evacuated from the at-risk area (Oldenborgh, 2018). In mid-September 2019, the amount of rainfall was extreme during Tropical Storm Imelda in Southeast Texas, USA, where over 1,000 people were affected by large-scale flooding and there were five deaths (Oldenborgh et al., 2019). Thus, it is necessary to assess how rainfall varies in each region of a country on a daily basis. Due to the climate pattern and meteorological conditions, Thailand is commonly separated into five regions: northern, northeastern, central, eastern, and southern. The rainfall in each region varies widely due to both location and seasonality. Importantly, Thailand’s rainfall data include many zeros with probability δ > 0 and positive right-skewed data following a lognormal distribution for the remainder of the probability. Thus, applying a delta-lognormal distribution (Aitchison, 1955) is appropriate.

The mean is a measure of the center of a set of observations (Casella & Berger, 2002) that can be used in statistical inference, while functions of the mean such as the ratio or difference between two means can also be used. These parameters have been applied in many research areas, such as medicine, fish stocks, pharmaceutics, and climatology. For example, they have been used for hypothesis testing of the effect of race on the average medical costs between African American and Caucasian patients with type I diabetes (Zhou, Gao & Hui, 1997), to estimate the mean charges for diagnostic tests on patients with unstable chronic medical conditions (Zhou & Tu, 2000; Tian, 2005; Tian & Wu, 2007; Li, Zhou & Tian, 2013), to estimate the maximum alcohol concentration in men in an alcohol interaction study (Tian & Wu, 2007; Krishnamoorthy & Oral, 2015), to estimate the mean red cod density around New Zealand as an indication of fish abundance (Fletcher, 2008; Wu & Hsieh, 2014), and to estimate the mean of the monthly rainfall totals to compare rainfall in Bloemfontein and Kimberley in South African (Harvey & van der Merwe, 2012).

In practice, the mean has been widely used in many fields, as mentioned before. When independent samples are recorded from several situations, then the common mean is of interest when studying more than one population. Many researchers have investigated methods for constructing confidence interval (CIs) for the common mean of several distributions. For example, Fairweather (1972) proposed a linear combination of Student’s t to construct CIs for the common mean of several normal distributions. Jordan & Krishnamoorthy (1996) solved the problem of CIs for the common mean under unknown and unequal variances based on Student’s t and independent F variables from several normal populations. Krishnamoorthy & Mathew (2003) presented the generalized CI (GCI) and compared it with the CIs constructed by Fairweather (1972), and Jordan & Krishnamoorthy (1996). Later, Lin & Lee (2005) developed a GCI for the common mean of several normal populations. Tian & Wu (2007) provided CIs for the common mean of several lognormal populations using the generalized variable approach, which was shown to be consistently better than the large sample (LS) approach. Lin & Wang (2013) studied the modification of the quadratic method to make inference via hypothesis testing and interval estimation for several lognormal means. Krishnamoorthy & Oral (2015) proposed the method of variance estimates recovery (MOVER) approach for the common mean of lognormal distributions.

As mentioned earlier, many researchers have developed CIs for the common mean of several normal and lognormal distributions. However, there has not yet been an investigation of statistical inference using the common mean of several delta-lognormal distributions. Since the common mean is used to study more than one population, the average precipitation in the five regions in Thailand can be estimated using it as there is an important need to estimate the daily rainfall trends in these regions. Furthermore, the daily rainfall records from the five regions in Thailand satisfy the assumptions for a delta-lognormal distribution. Herein, CIs for the common mean of several delta-lognormal models based on the fiducial GCI (FGCI), LS, MOVER, parametric bootstrap (PB), and highest posterior density (HPD) intervals based on Jeffreys’ rule (HPD-JR) and normal-gamma-beta (HPD-NGB) priors are proposed. The outline of this article is as follows. The ideas behind the proposed methods are detailed in the Methods section. Numerical computations are reported in ‘Simulation Studies and Results’. In ‘An Empirical Application’, the daily natural rainfall records of the five regions in Thailand are used to illustrate the efficacy of the methods. Finally, the paper is ended with a discussion and conclusions.

Methods

Let Wij = (Wi1, Wi2, …, Wini) be random samples drawn from a delta-lognormal distribution, for i = 1, 2, …, k and j = 1, 2, .., ni. There are three parameters in this distribution: the mean μi, variance σi2 and the probability of obtaining a zero observation δi. The distribution of Wij is given by (1) Hwij;μi,σi2,δi=δi;wij=0δi+1−δiGwij;μi,σi2;wij>0

where Gwij;μi,σi2 is a lognormal distribution function, denoted as LNμi,σi2 such that lnWij∼Nμi,σi2. The number of zeros has a binomial distribution ni0=#j:wij=0∼Bni,δi. The population mean of Wij is given by (2) ϑi=1−δiexpμi+σi22

The unbiased estimates of μi,σi2, and δi are μ ˆi=ni1−1∑j:wij>0 lnWij, σ ˆi2=ni1−1−1∑j:wij>0lnWij−μ ˆi2, and δ ˆi=ni0∕ni, respectively, where ni = ni(0) + ni(1); ni1=#j:wij>0. Suppose that the delta-lognormal mean in Eq. (2) for all k populations are the same, then according to Tian & Wu (2007) and Krishnamoorthy & Oral (2015), the common delta-lognormal mean is defined as (3) ϑ=1−δiexpμi+σi22.

For the ith sample, the estimates of ϑi are ϑ ˆi∗=1−δ ˆiexpμ ˆi+σ ˆi22 which contains the unbiased estimates μ ˆi, σ ˆi2 and δ ˆi. According to Longford (2009), the expected value of ϑ ˆi∗ is derived as

(4) Eϑ ˆi∗=1−Eδ ˆiEexpμ ˆi+σ ˆi22

(5) =1−δiexpμi+σi2ni1lili−σi2li∕2

where δ ˆi∼Nδi,δi1−δini as ni → ∞, Eexpμ ˆi= expμi+σi22ni1 and E[exp(ciYi)] = (1 − 2ci)−l∕2; Yi=liσ ˆi2σi2∼χli2 and ci=σi22li, σ ˆi2=ni1−1−1∑j=1ni1lnWij−μ ˆi2. If li−σi2li= exp−2σi2li12−12ni1, then we can obtain that

(6) Eϑ ˆi∗=1−δiexpμi+σi22ni1exp−2σi2li12−12ni1−li∕2=1−δiexpμi+σi22.

According to Aitchison & Brown (1963), the Aitchison estimate of ϑi is expressed as (7) ϑ ˆiAit=0;ni1=0wi1∕ni;ni1=11−δi ˆexpμi ˆψni1σ ˆi22;ni1>1

where ψa(b) is a Bessel function defined as (8) ψab=1+a−1ba+a−13a22!b2a+1+a−15a33!b3a+1a+3+...

To investigate the unbiased estimate ϑ ˆiAit, the expected value is

Eϑ ˆiAit= ∑j=1niPni1=jEϑi ˆ|ni1=j=0+Pni1=1Ewi1∕ni+∑j=2niPni1=jEϑi ˆ|ni1=j=Pni1=1expμi+σi22ni+∑j=2niPni1=jEni1ni expμi+σi22|ni1=j= ∑j=0niPni1=jEni1ni expμi+σi22|ni1=j=Eni1ni expμi+σi22=1−δiexpμi+σi22.

According to Shimizu & Iwase (1981), the uniformly minimum variance unbiased (UMVU) estimate of ϑi is (9) ϑ ˆiShi=0;ni1<1ni1ni expμ ˆi0F1ni1−12;ni1−14ni1Si2;ni1≥1

where Si2=∑j=1ni1lnWij−μ ˆi2 and 0F1a;z=∑m=0∞zmamm!; (10) am=1;m=0aa+1...a+m−1;m≥1

From Kunio (1983), E0F1ni1−12;a2Si2= expaσ2 is obtained, then (11) Eϑ ˆiShi=Eni1nexpμ ˆi0F1ni1−12,ni1−14ni1Si2=ni1−δini expμi+σi22ni1 expni1−12ni1σi2=1−δiexpμi+σi22

where E(ni(1)) = ni(1 − δi). The asymptotic variance of ϑ ˆiShi is given by

Varϑ ˆiShi= exp2μi+σi21ni2 ∑j=1ninij1−δijδni−jj2 expσi2j

0F1j−12;j−124j2σi4−1−δi2

(12) =exp2μi+σi2niδi1−δi+121−δi2σi2+σi4+On−2.

Actually, ψni1σ ˆi22=0F1ni1−12;ni1−14ni1Si2 such that ϑ ˆiShi and ϑ ˆiAit are the unbiased estimates of ϑi under different ideas, although their variances are the same i.e., Varϑ ˆiShi=Varϑ ˆiAit. Using μi ˆ,σ ˆi2, and δi ˆ from the samples, the estimated delta-lognormal mean ϑ ˆiAit and variance of ϑ ˆiAit are obtained. The following methods are the detailed construction of the CIs for the common delta-lognormal mean.

Fiducial generalized confidence interval

Fiducial inference was introduced by Fisher (1930). Fisher’s fiducial argument was used to develop a generalized fiducial recipe that could be extended to the application of fiducial ideas (Hannig, 2009). The concept of the fiducial interval has been advanced by the idea of the generalized pivotal quantity (GPQ) such that it is directly used to apply for generalized inference. Later, Hannig, Iyer & Patterson (2006) argued that a subclass of GPQs, the fiducial GPQ (FGPQ), provides a framework that shows the connection between a distribution and a parameter. Recall that μ ˆi∼Nμi,σi2∕ni1 and ni1−1σ ˆi2∕σi2∼χni1−12 are the independent random variables. The structure functions of μ ˆi and σ ˆi2 are (13) μ ˆi=μi+Viσi2ni1andσ ˆi2=σi2Uini1−1

which are the function of Vi and Ui, respectively, where Vi ∼ N(0, 1) and Ui∼χni1−12. Given the observed values, the estimates μ ˆi and σ ˆi2 can be obtained, and the unique solution of μ ˆi,σ ˆi2=μi+Viσi2ni1,σi2Uini1−1 becomes (14) μi=μ ˆi−Viσ ˆini1ni1−1Ui,σi2=ni1−1σ ˆi2Ui.

The respective FGPQs of μi and σi2 are (15) Gμi=μ ˆi−Vi∗σ ˆini1ni1−1Ui∗

(16) Gσi2=ni1−1σ ˆi2Ui∗

where Vi∗ and Ui∗ are independent copies of Vi and Ui, respectively. Hasan & Krishnamoorthy (2018) developed the FGPQ of δi using a beta distribution as Gδi′∼Betaαi,βi; αi = ni(1) + 0.5 and βi = ni(0) + 0.5. The FGPQ of ϑ based on k individual samples is (17) Gϑ=∑i=1kGwiGϑi ∑i=1kGwi

where Gϑi=Gδi′ expGμi+Gσi2∕2, Gwi=1∕GVarϑ ˆiAit, and GVarϑ ˆiAit= exp2Gμi+Gσi2Gδi′1−Gδi′+12Gδi′2Gσi2+Gσi4∕ni. Thus, the 100(1 − ζ)% FGCI for ϑ is (18) CIϑfgci=Lϑfgci,Uϑfgci=Gϑζ∕2,Gϑ1−ζ∕2

where Gϑ(ζ) denotes the ζth percentiles of Gϑ. Algorithm 1 shows the computational steps for obtaining the FGCI.

Algorithm 1: FGCI

(1) Generate Vi ∼ N(0, 1) and Ui∼χni1−12 are independent.

(2) Compute the FGPQs Gμi, Gσi2 and Gδi′.

(3) Compute Gwi and Gϑi leading to obtain Gϑ.

(4) Repeat steps 1-3, a number of times, m = 2500, compute 95%FGCI for ϑ, as given in Eq. (18).

Large sample interval

Recall that the Aitchitson estimator is ϑ ˆiAit=1−δi ˆexpμi ˆψni1σ ˆi2∕2 and the variance of ϑ ˆiAit is Varϑ ˆiAit= exp2μi+σi2δi1−δi+121−δi2σi2+σi4∕ni. The approximated variance is obtained by replacing μ ˆi, σ ˆi2 and δ ˆi. The pooled estimate of ϑi is given by (19) ϑ ˆ=∑i=1kwiϑ ˆiAit ∑i=1kwi

where wi=1∕Var ^ϑ ˆiAit. Hence, the 100(1 − ζ)% LS interval for ϑ is obtained as (20) CIϑls=Lϑls,Uϑls=ϑ ˆ−z1−ζ21∕ ∑i=1kwi,ϑ ˆ+z1−ζ21∕ ∑i=1kwi

where zζ denotes the ζth percentiles of standard normal N(0, 1). The LS interval can be estimated easily via ‘Algorithm 2’.

Algorithm 2: LS

(1) Compute ϑ ˆiAit and Var ^ϑ ˆiAit.

(2) Compute ϑ ˆ.

(3) Compute 95%LS interval for ϑ, as given in Eq. (20).

Method of variance estimates recovery

This method produces a closed-form CI that is easy to compute. For this reason, the MOVER CI for the common delta-lognormal mean is considered for k individual random samples. The MOVER for a linear combination of ϑi; i=1 , 2, …, k is as follows. Let ϑ ˆ1,ϑ ˆ2,…,ϑ ˆk be independent unbiased estimators of ϑ1, ϑ2, …, ϑk, respectively. In addition, let [li, ui] stand for the 100(1 − ζ)%CI for ϑi. According to Krishnamoorthy & Oral (2015), the 100(1 − ζ)%MOVER for ∑i=1kciϑi is given by (21) CI∑i=1kciϑi=L∑i=1kciϑi,U∑i=1kciϑi=∑i=1kciϑ ˆi−∑i=1kci2ϑ ˆi−li∗2,∑i=1kciϑ ˆi+∑i=1kci2ϑ ˆi−ui∗2

where li∗=li;ci>0ui;ci<0 and ui∗=ui;ci>0li;ci<0. Next, the closed-form CIs for ϑi are needed to construct MOVER for ϑ. Thus, ϑi is log-transformed as (22) lnϑi= lnδi∗+μi+σi2

where δi∗=1−δi. Let μ ˆi, and σ ˆi2 and δ ˆ∗ be the unbiased estimates of μi, σi2, and δi, respectively. The MOVER for a single delta-lognormal mean presented by Hasan & Krishnamoorthy (2018), the MOVER for ϑi is given by (23) Lϑi= explnδ ˆi∗+μ ˆi+σ ˆi2−lnδ ˆi∗−llnδi∗2+μ ˆi+σ ˆi2−lμi+σi22Uϑi= explnδ ˆi∗+μ ˆi+σ ˆi2−lnδ ˆi∗−ulnδi∗2+μ ˆi+σ ˆi2−uμi+σi22

where llnδi∗,ulnδi∗=lnδ ˆi∗+Ti,ζ∕222ni∓Ti,1−ζ∕2δ ˆi∗1−δ ˆi∗ni+Ti,ζ∕224ni2∕1+Ti,ζ∕22∕ni

(24) lμi+σi2,uμi+σi2=μ ˆi+σ ˆi2∕2−Zi,ζ∕2σ ˆi2ni12+σ ˆi441−ni1−1χi,1−ζ∕2,ni1−1221∕2,μ ˆi+σ ˆi2∕2+Zi,ζ∕2σ ˆi2ni12+σ ˆi44ni1−1χi,ζ∕2,ni1−12−121∕2.

Note that both Ti=ni1−niδ∗∕niδi∗1−δi∗∼dN0,1, and Zi=μ ˆi−μi∕σ ˆi2∕ni1∼dN0,1 are independent random variables. According to Krishnamoorthy & Oral (2015), the 100(1 − ζ)% MOVER interval for ϑ is (25) CIϑmover=Lϑ,Uϑ=∑i=1kwiϑ ˆiAit ∑i=1kwi− ∑i=1kwi2ϑ ˆiAit−Lϑi2 ∑i=1kwi2,∑i=1kwiϑ ˆiAit ∑i=1kwi− ∑i=1kwi2ϑ ˆiAit−Uϑi2 ∑i=1kwi2

where wi=1∕Var ^ϑ ˆiAit. ‘Algorithm 3’ describes the steps to construct the MOVER interval.

Algorithm 3: MOVER

(1) Compute CIs for lnδi∗ and μi+σi2 are llnδi∗,ulnδi∗ and lμi+σi2,uμi+σi2, respectively.

(2) Compute MOVER for ϑi, as given in Eq. (23).

(3) Compute 95%MOVER for ϑ, given in Eq. (25).

Parametric Bootstrap

This is developed from the parametric bootstrap on the common mean of several heterogeneous log-normal distributions, proposed by Malekzadeh & Kharrati-Kopaei (2019). The delta-lognormal mean is transformed by taking the logarithm as (26) μi= lnϑ1−δi−σi22.

The likelihood of ϑ,σi2,δi is (27) Lϑ,σi2,δi|wij= ∏i=1knini0δi1−δi12πσi2ni1∕2 exp−12σi2 ∑j=1ni1lnwij− lnϑ1−δi+σi222

which enables obtaining the maximum likelihood estimates of lnϑ and σi2 as (28) lnϑ ˆmle=∑i=1kw ˆmle,iμ ˆi+ ln1−δ ˆi+N∕2∑i=1kw ˆmle,iσ ˆmle,i2=−2+21+σ ˆi2+μ ˆ−lnϑ ˆ∕1−δ ˆi2

where w ˆmle,i=ni1∕σ ˆmle,i2 and lnϑ ˆ=∑i=1kw ˆiμ ˆi+ln1−δ ˆi+N∕2∑i=1kw ˆi; w ˆi=ni1∕σ ˆi2. If δi = 0, then it becomes the common lognormal mean (see Krishnamoorthy & Oral (2015) for a detailed explanation). By applying central limit theorem, we obtain lnϑ ˆmle− lnϑ∑i=1kw ˆmle,i∼N0,1 such that T=lnϑ ˆmle− lnϑ2∑i=1kw ˆmle,i∼χni1−12. It is well-known that μ ˆi, σ ˆi2 and δ ˆi are independent random variables for which μ ˆi∼Nlnϑ1−δi−σi22,σi2∕ni1, ni1−1σ ˆi2∕σi2∼χni1−12 and δ ˆi∼Nδ,δ1−δ∕ni are obtained, respectively. Let η=μi+σi2∕2 so that we can write T=∑i=1kw ˆmle,iμ ˆi+ln1−δ ˆi−η− ln1−δi+N∕2∑i=1kw ˆmle,i. It can be seen that the distribution of T is complicated, possibly depending on nuisance parameters σi2 and δi, but not on lnϑ. Thus, the exact distribution of T is unknown in practice, and so we propose the PB pivotal variable corresponding to TPB as (29) TPB=lnϑ ˆmlePB− lnϑ ˆ2 ∑i=1kw ˆmle,iPB

where lnϑ ˆmlePB=∑i=1kw ˆmle,iPBμ ˆiPB+ln1−δ ˆiPB+N∕2∑i=1kw ˆmle,iPB, w ˆiPB=ni1∕σ ˆi2B, μ ˆiPB∼Nμ ˆiB,σ ˆi2B∕ni1, σ ˆi2PB∼σ ˆiB2χni1−12∕ni1−1 and δ ˆPB∼betani0B+0.5,ni1B+0.5, ni0B=niδ ˆiB, and ni1B=ni−ni0B. Note that μ ˆiB, σ ˆi2B, and δ ˆiB are the observed values of μ ˆi, σ ˆi2, and δ ˆi, respectively, from random sampling with replacement based on the bootstrap approach. Thus, the 100(1 − ζ)% PB interval for ϑ is given by (30) CIϑpb= explnϑ ˆmle∓qζPB∕ ∑i=1kw ˆmle,i

where qζPB denotes the (1 − ζ)th percentile of distribution of TPB. The PB interval can be constructed as shown in ‘Algorithm 4’.

Algorithm 4: PB

(1) Compute μ ˆi, σ ˆi2 and δ ˆ leading to obtain lnϑ ˆ.

(2) Compute lnϑ ˆmle and σ ˆmle,i2.

(3) Generate μ ˆiPB, σ ˆi2PB and δ ˆiPB leading to compute lnϑ ˆmlePB.

(4) Repeat steps 1-3, a number of time m = 2500, compute TPB to obtain qζPB.

(5) Compute 95%PB interval for ϑ, as given in Eq. (30).

Highest posterior density intervals

The HPD interval is constructed from the posterior distribution, as defined by Box & Tiao (1973). Note that the prior of ϑi is updated with its likelihood function thereby obtaining the posterior distribution based on the Bayesian approach. Recall that Wij∼Δμi,σi2,δi, then the likelihood is given by (31) Pwij|μi,σi2,δi∝∏i=1kδini01−δini1σi2−ni1∕2 exp−12σi2 ∑j=1ni1lnwij−μi2.

For k individual samples, Miroshnikov, Wei & Conlon (2015) described the pooled independent sub-posterior samples toward the joint posterior distributions ϑ are combined using weighted averages as follows: (32) ϑpost= ∑i=1kwiϑipost∑i=1kwi−1

where ϑipost are the posterior samples of ϑi, for i = 1, 2, …, k. The inverse of the sample variance is used to weight the posterior based on the ith samples is denoted as wi=Var−1ϑ ˆi|wij. Different priors have been developed for estimating the common delta-lognormal mean, two of which are derived in the following subsections.

Jeffreys’ rule prior

Harvey & van der Merwe (2012) defined this prior as (33) PϑJR∝∏i=1kσi−3δi−1∕21−δi1∕2

which is combined with the likelihood Eq. (34) to obtain the posterior of ϑ as (34) Pwij|ϑ∝∏i=1kδini0−1∕21−δini1+1∕2σi2−ni1+3∕2 exp−12σi2 ∑j=1ni1lnwij−μi2∝∏i=1kδini0+1∕2−11−δini1+3∕2−1σi2−ni1+12−1 exp−12σi2ni1−1σ ˆi2+ni1μ ˆi−μi2.

This leads to obtaining the marginal posterior distributions of μi, σi2 and δi as (35) μiJR|σi,JR2,wij∼Nμ ˆi,σi2JR∕ni1σi2JR|wij∼IGni1+1∕2,ni1+1σ ˆi2∕2δiJR|wij∼betani0+1∕2,ni1+3∕2.

The pooled posterior of ϑ is weighted by its inversely estimated variance as follows: (36) ϑpost= ∑i=1kwiJRϑiJRp∑i=1kwiJR−1

where

ϑiJRp=1−δiJRexpμiJR+σi2JR∕2

wiJR=ni−1 exp2μiJR+σi2JRδiJR1−δiJR+121−δiJR2σi2JR+σi4JR−1.

From Eq. (36), the 100(1 − ζ)%HPD-based Jeffreys’ rule prior (HPD-JR) for ϑ is constructed as follows:

Normal-gamma-beta prior

Maneerat, Niwitpong & Niwitpong (2020) proposed a HPD based on the normal-gamma prior for the ratio of delta-lognormal variances that worked better than the HPD-JR of Harvey & van der Merwe (2012). Suppose that Y = lnW be a random variable of normal distribution with mean μ = (μ1, μ2, …, μk) and precision λ = (λ1, λ2, …, λk) where W ∼ LN(μ, λ) and λi=σi−2. The HPD-based normal-gamma-beta prior (HPD-NGB) of ϑ = (μi, λi, δi)′ is defined as (37) Pϑ∝∏i=1kλi−1δi1−δi−1∕2

where (μi, λi) follows a normal-gamma distribution, and δi follows a beta distribution, denoted as (μi, λi) ∼ NG(μi, λi|μ, ki(0) = 0, αi(0) =  − 1∕2, βi(0) = 0) and δi ∼ beta(1∕2, 1∕2), respectively. When the the prior Eq. (37) is combined with the likelihood Eq. (34), then the posterior density of ϑ becomes (38) Pϑ|wij∝∏i=1kδini0−1∕21−δini1−1∕2λini1−12−1 exp−λi2∑j=1ni1lnwij−μ ˆi2λi1∕2 exp−ni1λi2μi−μi∗2

which can be integrated out to obtain the marginal posterior distributions of μi, λi and δi as follows: (39) μiNGB|wij∼tdfμi|μ ˆi,∑j=1ni1lnwij−μ ˆi2∕ni1ni1−1λiNGB|wij∼Gλi|ni1−1∕2,∑j=1ni1lnwij−μ ˆi2∕2δiNGB|wij∼betani0+1∕2,ni1+1∕2

where df = 2 (ni(1) − 1) and σi2NGB|wij∼IGσi2|ni1−1∕2,∑j=1ni1lnwij−μ ˆi2∕2. Similarly, the pooled posterior of ϑ is given by (40) ϑpost= ∑i=1kwiNGBϑiNGBp∑i=1kwiNGB−1

where ϑiNGBp=1−δiNGBexpμiNGB+σi2NGB∕2wiNGB =ni−1 exp2μiNGB+σi2NGBδiNGB1−δiNGB121−δiNGB2σi2NGB+σi4NGB−1.

Hence, the 100(1 − ζ)%HPD-HGB for ϑ is constructed in Eq. (40). ‘Algorithm 5’ details the steps to construct the HPD-JR and HPD-NGB.

Algorithm 5: HPD-JR and HPD-NGB

(1) Compute μ ˆi, σ ˆi2 and δ ˆ.

(2) Generate the posterior densities of μi, σi2 and δi based-Jeffreys’ rule (JR) and normal-gamma-beta (NGB) priors, as given in Eq. (35) and Eq. (39), respectively.

(3) Compute the pooled posterior of ϑ based on JR and NGB priors, as given in Eq. (36) and Eq. (40), respectively.

(4) Compute 95%HPD-JR and HPD-NGB for ϑ, defined by Box & Tiao (1973).

Simulation Studies and Results

The performances of the CIs were assessed by comparing their coverage probabilities (CPs) and average length (ALs) using Monte Carlo simulation. The best-performing CI is the one where the CP is closest to or greater than the nominal confidence level 1 − ζ while also having an AL with the narrowest width. The CIs for the common delta-lognormal mean constructed using FGCI, LS, MOVER, PB, HPD-JR, and HPD-NGB were assessed in the study, the parameter settings for which are provided in Table 1. The number of generated random samples was fixed at M = 5000. For FGCI, the number of FGPQs was Q = 2500 for each set of 5,000 random samples. ‘Algorithm 6’ shows the computational steps to estimate the CP and AL performances of all of the methods.

Table 1 Parameter settings for sample cases k = 2, 5, 10.

Scenarios	(n1, …, nk)	(δ1, …, δk)	σ12,…,σk2	
k = 2	
1–9	(302)	(0.1,0.2), (0.2,0.5), (0.3,0.7)	(1,2), (2,4), (3,5)	
10–18	(30,50)	(0.1,0.2), (0.2,0.5), (0.3,0.7)	(1,2), (2,4), (3,5)	
19–27	(502)	(0.1,0.2), (0.2,0.5), (0.3,0.7)	(1,2), (2,4), (3,5)	
28–36	(50,100)	(0.1,0.2), (0.2,0.5), (0.3,0.7)	(1,2), (2,4), (3,5)	
37–45	(1002)	(0.1,0.2), (0.2,0.5), (0.3,0.7)	(1,2), (2,4), (3,5)	
k = 5	
46–54	(305)	(0.05, 0.12, 0.22), (0.22, 0.43), (0.52, 0.73)	(12, 23), (22, 33), (32, 53)	
55–63	(302, 503)	(0.05, 0.12, 0.22), (0.22, 0.43), (0.52, 0.73)	(12, 23), (22, 33), (32, 53)	
64–72	(302, 502, 100)	(0.05, 0.12, 0.22), (0.22, 0.43), (0.52, 0.73)	(12, 23), (22, 33), (32, 53)	
73–81	(30, 502, 1002)	(0.05, 0.12, 0.22), (0.22, 0.43), (0.52, 0.73)	(12, 23), (22, 33), (32, 53)	
82–90	(505)	(0.05, 0.12, 0.22), (0.22, 0.43), (0.52, 0.73)	(12, 23), (22, 33), (32, 53)	
91–99	(502, 1003)	(0.05, 0.12, 0.22), (0.22, 0.43), (0.52, 0.73)	(12, 23), (22, 33), (32, 53)	
100–108	(1005)	(0.05, 0.12, 0.22), (0.22, 0.43), (0.52, 0.73)	(12, 23), (22, 33), (32, 53)	
k = 10	
109–114	(305, 505)	(0.15, 0.25), (0.25, 0.55)	(15, 25), (25, 45), (35, 55)	
115–120	(303, 503, 1004)	(0.15, 0.25), (0.25, 0.55)	(15, 25), (25, 45), (35, 55)	
121–126	(505, 1005)	(0.15, 0.25), (0.25, 0.55)	(15, 25), (25, 45), (35, 55)	
Notes.

Note: (305) stands for (30, 30, 30, 30, 30).

Table 2 Performance measures of 95%CIs for ϑ: 2 sample cases.

Scenarios	CP	AL	
	FG	LS	MO	PB	HJ	HN	FG	LS	MO	PB	HJ	HN	
k = 2	
1	0.959	0.897	0.967	0.994	0.916	0.941	1.556	1.296	2.005	2.324	1.353	1.436	
2	0.958	0.857	0.947	0.996	0.924	0.941	5.169	3.770	7.287	8.631	4.186	4.335	
3	0.963	0.821	0.959	0.996	0.919	0.932	13.088	8.675	23.312	22.883	9.905	10.220	
4	0.962	0.886	0.978	0.995	0.917	0.939	1.487	1.211	2.181	2.155	1.247	1.386	
5	0.953	0.832	0.962	0.995	0.913	0.922	4.875	3.487	9.881	7.818	3.811	4.066	
6	0.951	0.793	0.971	0.991	0.901	0.912	12.311	7.740	37.615	21.129	8.875	9.378	
7	0.961	0.829	0.972	0.982	0.920	0.940	1.511	1.095	3.968	2.173	1.224	1.406	
8	0.950	0.778	0.974	0.995	0.900	0.911	4.821	3.123	293.620	7.649	3.566	3.916	
9	0.939	0.725	0.973	0.988	0.866	0.887	13.159	7.067	8.0e4	23.632	8.680	9.419	
10	0.960	0.900	0.965	0.992	0.915	0.941	1.503	1.249	1.936	2.225	1.362	1.395	
11	0.961	0.848	0.941	0.992	0.924	0.940	5.128	3.712	6.765	8.667	4.298	4.368	
12	0.965	0.819	0.952	0.998	0.919	0.931	12.297	8.382	20.057	21.597	9.819	9.894	
13	0.960	0.896	0.977	0.992	0.917	0.942	1.366	1.147	1.909	2.004	1.203	1.271	
14	0.961	0.851	0.964	0.996	0.916	0.931	4.593	3.422	7.236	7.458	3.761	3.889	
15	0.949	0.790	0.958	0.994	0.894	0.905	11.116	7.517	22.293	19.310	8.507	8.718	
16	0.963	0.860	0.972	0.974	0.928	0.943	1.354	1.033	2.141	1.928	1.155	1.257	
17	0.952	0.803	0.976	0.992	0.900	0.917	4.397	3.048	10.772	6.889	3.418	3.630	
18	0.940	0.737	0.968	0.989	0.872	0.889	11.065	6.663	43.755	19.011	7.903	8.247	
19	0.961	0.914	0.966	0.992	0.921	0.946	1.153	1.009	1.382	1.696	1.043	1.076	
20	0.965	0.895	0.946	0.991	0.938	0.949	3.668	2.924	4.309	5.981	3.178	3.229	
21	0.962	0.863	0.952	0.996	0.930	0.940	8.747	6.665	11.805	14.651	7.272	7.395	
22	0.958	0.910	0.978	0.985	0.919	0.944	1.091	0.945	1.414	1.555	0.945	1.031	
23	0.965	0.883	0.969	0.996	0.926	0.937	3.336	2.695	4.578	5.204	2.811	2.950	
24	0.961	0.840	0.972	0.995	0.921	0.928	7.887	5.987	13.164	12.757	6.338	6.605	
25	0.969	0.868	0.980	0.958	0.930	0.953	1.120	0.866	1.610	1.503	0.937	1.070	
26	0.954	0.839	0.970	0.997	0.916	0.926	3.208	2.433	6.544	4.735	2.621	2.830	
27	0.946	0.773	0.970	0.992	0.893	0.903	7.803	5.443	26.105	12.382	6.011	6.376	
28	0.958	0.912	0.972	0.979	0.916	0.947	1.119	0.952	1.397	1.615	1.054	1.051	
29	0.956	0.872	0.921	0.958	0.927	0.943	3.745	2.836	4.238	6.098	3.338	3.330	
30	0.961	0.846	0.937	0.987	0.925	0.936	8.488	6.274	10.833	13.991	7.332	7.320	
31	0.962	0.927	0.985	0.978	0.919	0.949	0.984	0.876	1.322	1.433	0.908	0.929	
32	0.960	0.880	0.958	0.992	0.925	0.940	3.214	2.618	4.169	5.150	2.818	2.860	
33	0.958	0.838	0.960	0.994	0.910	0.925	7.360	5.744	10.824	12.105	6.256	6.279	
34	0.963	0.888	0.977	0.922	0.938	0.954	0.975	0.785	1.322	1.352	0.876	0.922	
35	0.958	0.860	0.971	0.995	0.917	0.929	2.915	2.343	4.321	4.424	2.486	2.586	
36	0.951	0.820	0.973	0.995	0.901	0.916	6.726	5.103	11.951	10.823	5.511	5.626	
37	0.957	0.935	0.960	0.970	0.927	0.948	0.802	0.722	0.923	1.168	0.743	0.753	
38	0.955	0.916	0.926	0.953	0.942	0.948	2.442	2.044	2.541	3.935	2.220	2.219	
39	0.957	0.888	0.939	0.981	0.937	0.945	5.608	4.594	6.295	9.049	4.984	4.998	
40	0.961	0.942	0.975	0.957	0.924	0.954	0.740	0.679	0.911	1.062	0.659	0.702	
41	0.961	0.920	0.960	0.988	0.933	0.950	2.199	1.925	2.558	3.401	1.958	2.012	
42	0.955	0.875	0.960	0.994	0.925	0.931	4.976	4.209	6.298	7.813	4.318	4.439	
43	0.967	0.909	0.980	0.863	0.937	0.960	0.773	0.625	0.972	1.012	0.659	0.743	
44	0.960	0.896	0.970	0.993	0.928	0.939	2.076	1.750	2.684	3.013	1.788	1.921	
45	0.952	0.835	0.970	0.996	0.908	0.914	4.683	3.786	7.007	7.008	3.952	4.182	
Notes.

FG fiducial generalized confidence interval

MO method of variance estimates

HJ HPD-based Jeffreys’ rule prior

HPD-JR HN, HPD-based normal-gamma-beta prior

Bold denoted as the best-performing method each case.

Figure 1 CP performances of 95%CIs for ϑ: 2 sample cases in the following cases (sample sizes, variances): (A) (302, 1, 2), (B) (30, 30, 2, 4), (C) (30, 30, 3, 5), (D) (30, 50, 1, 2), (E) (30, 50, 2, 4), (F) (30, 50, 3, 5), (G) (502, 1, 2), (H) (502, 2, 4), (I) (502, 3, 5), (J) (50, 100, 1, 2), (K) (50, 100, 2, 4), (L) (50, 100, 3, 5), (M) (1002, 1, 2), (N) (1002, 2, 4), (O) (1002, 3, 5).

Algorithm 6: Comparison of CPs and ALs for all CIs

(1) For g = 1 to M. Generate wij∼Δμi,σi2,δi.

(2) Compute the unbiased estimates μ ˆi, σ ˆi2 and δ ˆ.

(3) Compute the 95%CIs for ϑ based on FGCI, LS, MOVER, PB and the HPDs via Algorithm 1, 2, 3, 4 and 5, respectively.

(4) Let Ag = 1 if ϑ falls within the intervals of FGCI, LS, MOVER, PB or the HPDs, else Ag = 0.

(5) The CP and AL for each method are obtained by CP=1∕M∑g=1MAg and AL = (U − L)∕M, respectively, where U and L are the upper and lower confidence limits, respectively. (end g loop)

The numerical results for the CI performances are presented in terms of CP and AL for various sample cases. For k = 2 (Table 2 and Fig. 1), FGCI performed well for small-to-moderate sample sizes, as well as for large σi2 and a moderate-to-large sample size. HPD-NGB attained stable and the best CP and AL values for small σi2 and a moderate-to-large sample size. MOVER and PB attained correct CPs but wider ALs than the other methods whereas LS and HPD-JR had lower CPs and narrower ALs. For k = 5 (Table 3 and Fig. 2), there were only two methods producing better CPs than the other methods in the various situations: MOVER (small δi and σi2) and PB (large δi and σi2). Moreover, the results were similar for k = 10 (Table 4 and Fig. 3).

Figure 2 CP performances of 95%CIs for ϑ: 5 sample cases in the following cases (sample sizes, variances): (A) (305, 12, 23), (B) (305, 22, 33), (C) (305, 32, 53), (D) (302, 503, 12, 23), (E) (302, 503, 22, 33), (F) (302, 503, 32, 53), (G) (302, 503, 100, 12, 23), (H) (302, 503, 100, 22, 33), (I) (302, 503, 100, 32, 53), (J) (30, 502, 1002, 12, 23), (K) (30, 502, 1002, 12, 23), (L) (30, 502, 1002, 12, 23), (M) (505, 12, 23), (N) (505, 22, 33), (O) (505, 32, 53), (P) (502, 1003, 12, 23), (Q) (502, 1003, 22, 33), (R) (502, 1003, 32, 53), (S) (1005, 12, 23), (T) (1005, 22, 33), (U) (1005, 32, 53).

Table 3 Performance measures of 95% CIs for ϑ: 5 sample cases.

Scenarios	CP	AL	
	FG	LS	MO	PB	HJ	HN	FG	LS	MO	PB	HJ	HN	
k = 5	
46	0.885	0.790	0.988	0.989	0.757	0.846	0.963	0.819	1.794	1.532	0.848	0.956	
47	0.789	0.627	0.973	0.996	0.674	0.715	2.240	1.908	4.982	3.897	1.991	2.176	
48	0.840	0.613	0.953	0.997	0.723	0.746	5.325	4.529	13.769	12.250	4.744	4.870	
49	0.894	0.800	0.993	0.978	0.779	0.864	0.900	0.765	1.825	1.439	0.773	0.905	
50	0.783	0.623	0.972	0.998	0.680	0.711	2.008	1.711	5.203	3.608	1.750	1.955	
51	0.797	0.580	0.959	0.996	0.680	0.701	4.700	4.066	16.626	11.353	4.118	4.287	
52	0.893	0.735	0.989	0.896	0.816	0.853	0.753	0.589	2.849	1.433	0.636	0.764	
53	0.768	0.517	0.977	0.997	0.666	0.676	1.474	1.168	19.967	3.364	1.282	1.406	
54	0.742	0.467	0.983	0.996	0.624	0.629	3.250	2.654	1.5e4	11.238	2.817	2.855	
55	0.884	0.779	0.988	0.979	0.743	0.846	0.940	0.777	1.739	1.434	0.857	0.930	
56	0.806	0.645	0.973	0.995	0.681	0.740	2.204	1.822	4.586	3.561	2.045	2.141	
57	0.858	0.622	0.949	0.986	0.725	0.771	5.620	4.542	12.575	12.073	5.122	5.162	
58	0.901	0.827	0.995	0.962	0.770	0.870	0.845	0.728	1.699	1.326	0.771	0.841	
59	0.793	0.644	0.978	0.997	0.675	0.726	1.904	1.629	4.351	3.262	1.750	1.850	
60	0.825	0.605	0.952	0.997	0.710	0.734	4.753	4.058	12.745	10.793	4.373	4.353	
61	0.905	0.785	0.992	0.822	0.809	0.865	0.685	0.564	1.632	1.219	0.620	0.686	
62	0.786	0.578	0.969	0.993	0.683	0.704	1.368	1.142	4.477	2.775	1.260	1.309	
63	0.755	0.496	0.963	0.998	0.639	0.637	3.177	2.714	18.995	8.911	2.884	2.822	
64	0.892	0.787	0.991	0.970	0.737	0.858	0.928	0.751	1.740	1.364	0.872	0.919	
65	0.822	0.647	0.975	0.996	0.673	0.763	2.168	1.738	4.371	3.326	2.047	2.114	
66	0.852	0.593	0.943	0.981	0.715	0.767	5.710	4.413	12.195	11.422	5.267	5.278	
67	0.905	0.827	0.996	0.949	0.768	0.873	0.816	0.697	1.637	1.256	0.770	0.811	
68	0.801	0.654	0.979	0.995	0.683	0.737	1.839	1.549	4.069	3.016	1.753	1.797	
69	0.821	0.595	0.947	0.994	0.693	0.733	4.806	3.976	12.174	10.326	4.431	4.432	
70	0.917	0.803	0.994	0.775	0.817	0.886	0.650	0.539	1.499	1.133	0.616	0.650	
71	0.804	0.612	0.973	0.992	0.692	0.730	1.310	1.094	3.962	2.543	1.236	1.262	
72	0.756	0.502	0.958	0.997	0.631	0.646	3.158	2.695	16.604	8.356	2.888	2.835	
73	0.924	0.832	0.994	0.942	0.772	0.893	0.822	0.673	1.505	1.186	0.856	0.808	
74	0.853	0.699	0.985	0.990	0.696	0.798	1.971	1.589	3.823	2.899	2.000	1.923	
75	0.883	0.652	0.952	0.945	0.755	0.817	5.330	4.072	9.997	9.911	5.224	4.974	
76	0.924	0.857	0.997	0.913	0.771	0.901	0.723	0.626	1.418	1.088	0.746	0.715	
77	0.826	0.695	0.986	0.989	0.689	0.767	1.670	1.406	3.476	2.610	1.692	1.632	
78	0.854	0.638	0.955	0.984	0.718	0.771	4.456	3.628	9.715	8.788	4.311	4.160	
79	0.930	0.846	0.998	0.683	0.811	0.900	0.581	0.486	1.253	0.964	0.586	0.580	
80	0.830	0.658	0.981	0.980	0.705	0.762	1.215	1.019	3.179	2.168	1.225	1.181	
81	0.788	0.555	0.967	0.997	0.675	0.689	2.992	2.554	11.873	7.026	2.927	2.738	
82	0.915	0.844	0.993	0.964	0.788	0.889	0.769	0.662	1.337	1.158	0.692	0.753	
83	0.858	0.735	0.982	0.993	0.741	0.804	1.882	1.599	3.605	2.920	1.698	1.825	
84	0.886	0.705	0.969	0.981	0.782	0.827	4.650	3.895	8.767	9.068	4.208	4.335	
85	0.925	0.865	0.998	0.939	0.803	0.897	0.707	0.618	1.315	1.068	0.618	0.700	
86	0.834	0.705	0.987	0.994	0.735	0.775	1.683	1.439	3.493	2.683	1.482	1.642	
87	0.855	0.684	0.968	0.994	0.751	0.783	4.027	3.489	8.924	8.068	3.613	3.766	
88	0.929	0.824	0.994	0.677	0.835	0.903	0.611	0.495	1.322	0.993	0.515	0.616	
89	0.823	0.627	0.981	0.985	0.729	0.749	1.284	1.045	3.692	2.296	1.121	1.250	
90	0.799	0.578	0.972	0.997	0.699	0.705	2.875	2.453	13.603	6.644	2.519	2.641	
91	0.927	0.831	0.997	0.906	0.777	0.898	0.753	0.614	1.389	1.064	0.703	0.735	
92	0.871	0.731	0.988	0.986	0.720	0.820	1.821	1.466	3.459	2.601	1.721	1.769	
93	0.905	0.693	0.957	0.897	0.791	0.852	5.015	3.768	8.461	8.829	4.621	4.690	
94	0.931	0.879	0.999	0.873	0.781	0.909	0.651	0.571	1.279	0.972	0.608	0.639	
95	0.847	0.738	0.991	0.986	0.719	0.797	1.541	1.313	3.117	2.351	1.447	1.499	
96	0.875	0.679	0.966	0.969	0.760	0.806	4.125	3.374	8.002	7.707	3.808	3.865	
97	0.935	0.866	0.998	0.541	0.832	0.911	0.529	0.450	1.097	0.856	0.493	0.523	
98	0.848	0.697	0.986	0.971	0.735	0.782	1.126	0.956	2.572	1.916	1.060	1.091	
99	0.817	0.613	0.963	0.994	0.698	0.725	2.784	2.418	7.510	6.042	2.565	2.571	
100	0.941	0.888	0.998	0.863	0.813	0.920	0.557	0.484	0.954	0.806	0.510	0.536	
101	0.906	0.827	0.995	0.973	0.799	0.875	1.413	1.201	2.515	2.029	1.288	1.361	
102	0.929	0.790	0.975	0.861	0.845	0.889	3.639	2.946	5.529	6.174	3.365	3.428	
103	0.948	0.923	1.000	0.801	0.816	0.931	0.501	0.456	0.909	0.741	0.452	0.487	
104	0.888	0.816	0.996	0.978	0.784	0.853	1.253	1.095	2.373	1.852	1.121	1.216	
105	0.905	0.775	0.981	0.953	0.822	0.859	3.147	2.678	5.326	5.441	2.893	2.975	
106	0.955	0.907	0.999	0.289	0.852	0.943	0.438	0.372	0.838	0.668	0.373	0.433	
107	0.881	0.761	0.994	0.939	0.781	0.833	0.992	0.823	2.044	1.536	0.863	0.972	
108	0.868	0.722	0.984	0.987	0.781	0.805	2.331	2.005	5.072	4.308	2.088	2.208	
Notes.

FG fiducial generalized confidence interval

MO method of variance estimates

HJ HPD-based Jeffreys’ rule prior

HPD-JR HN, HPD-based normal-gamma-beta prior

Bold denoted as the best-performing method each case.

Figure 3 CP performances of 95%CIs for ϑ: 10 sample cases in the following cases (sample sizes, variances): (A) (305, 505, 12, 25), (B) (305, 505, 25, 45), (C) (305, 505, 35, 55), (D) (303, 503, 1004, 12, 25), (E) (303, 503, 1004, 25, 45), (F) (303, 503, 1004, 35, 55), (G) (505, 1005, 15, 25), (H) (505, 1005, 25, 45), (I) (505, 1005, 32, 53).

Table 4 Performance measures of 95% CIs for ϑ: 10 sample cases.

	CP	AL	
Scenarios	FG	LS	MO	PB	HJ	HN	FG	LS	MO	PB	HJ	HN	
k = 10	
109	0.728	0.675	0.998	0.927	0.566	0.692	0.612	0.501	1.554	0.932	0.545	0.623	
110	0.661	0.500	0.979	0.891	0.570	0.588	1.644	1.291	3.867	3.278	1.500	1.637	
111	0.504	0.352	0.950	0.978	0.481	0.404	3.159	2.561	8.645	7.286	2.996	3.076	
112	0.720	0.692	0.999	0.904	0.587	0.690	0.557	0.459	1.519	0.832	0.483	0.574	
113	0.532	0.452	0.976	0.985	0.512	0.462	1.393	1.159	3.853	2.682	1.260	1.404	
114	0.361	0.290	0.955	0.998	0.403	0.274	2.556	2.218	8.570	5.943	2.411	2.505	
115	0.789	0.723	0.999	0.808	0.561	0.762	0.554	0.440	1.416	0.789	0.546	0.560	
116	0.716	0.524	0.985	0.578	0.590	0.653	1.635	1.180	3.478	2.915	1.559	1.624	
117	0.593	0.406	0.964	0.872	0.519	0.507	3.289	2.406	7.754	6.380	3.189	3.214	
118	0.782	0.773	1.000	0.780	0.586	0.758	0.477	0.404	1.317	0.696	0.474	0.483	
119	0.626	0.514	0.988	0.947	0.535	0.561	1.337	1.076	3.348	2.360	1.284	1.341	
120	0.447	0.347	0.965	0.992	0.450	0.355	2.570	2.108	7.290	5.180	2.506	2.531	
121	0.826	0.773	1.000	0.736	0.592	0.796	0.488	0.399	1.266	0.695	0.444	0.486	
122	0.774	0.620	0.994	0.438	0.647	0.720	1.460	1.086	3.072	2.512	1.328	1.438	
123	0.659	0.460	0.977	0.798	0.553	0.577	3.002	2.236	6.597	5.502	2.775	2.921	
124	0.828	0.826	1.000	0.708	0.606	0.802	0.426	0.368	1.187	0.615	0.387	0.427	
125	0.688	0.595	0.995	0.912	0.591	0.627	1.205	0.992	2.912	2.039	1.094	1.197	
126	0.520	0.426	0.979	0.984	0.486	0.439	2.390	1.989	6.222	4.479	2.224	2.344	
Notes.

FG fiducial generalized confidence interval

MO method of variance estimates recovery

HJ HPD-based Jeffreys’ rule prior

HPD-JR HN, HPD-based normal-gamma-beta prior

Bold denoted as the best-performing method each case.

As previously mentioned, our findings show that FGCI works well for small sample case because the FGPQ of σi2 might contain some weak points that affect the FGPQ of μi as the sample case increases. For large sample sizes, MOVER was the best method for small σ2, which is possibly caused by the CI for μi+σi2. Meanwhile, the next best one was PB, which has the strong point of using a resampling technique to collect information about several populations even when the variance σ2 is large.

Table 5 Daily rainfall data in five Thailand’s regions on August 5, 2019.

Northern	Northeastern	Central	Eastern	Southern	
3	0	3	0	0	49.5	0	0	0	2.9	3.2	0	4.1	0	0	2.7	
2.6	5	0	40	1.5	10.5	0	0	0	0.2	0	3.2	0	0	0	0	
1	23.8	0	3.5	18.5	60.4	4	0	11	0.3	0	10.4	11.5	3.5	0	0	
3.6	16	0	0	42	12.7	0	0	0	2.5	4.7	1.1	2.5	13.6	0	0	
0	11.5	0	12	9.1	6.8	0	20.3	0	0.4	19.3	0.2	9.7	0	0.2	0	
13.2	1.2	0	15	6	69.3	0	0	0	0.4	3.1	4.3	10.4	0	0	0	
22.4	10.3	0	0	7.5	36.5	0	2.4	0.3	1.1	2.9	0	9.6	0	0	0	
1.4	1.7		0	1.5	0	8.6	0	0	1	0	5.7	0 19	0	0	0	
18.3	5.5	0	0.7	6.3	0	0	0	0	1.3	0.9	0	8.3	0	0	0	
0	7.3	0	0	0	0	0	0	0	0.1	0	0	0	4.8	0	6.2	
15.5	24.3	1.7	3	0.4	0	0	0	0	2.9	0	0.2	0	0	0	0	
0	27.2	2.3	0	0	3.8	0	0	0	0	2.6	0.1	0	0	0	0	
0	12.6	0.5	0	0	0	0	3.2	0	1	17	62.8	0	0	0	6.1	
0	22.7	3.9	0	0	0	0	0	0	4.7	0	36.7	17.8	0	0	0	
9.8	0	6.9	29.4	1.8	0	0	0	0	0.5	3.5	15.6	12.3	0	0	0	
24.3	2.6	2.2	48	0	0	0	0	0	5	0	50	2.5	0	0	0	
24.6	0	3.2	0	0	0	6	0	0	2.5	0	35.5	0	0	0	0.3	
8.8	3.2	5.3	70.8	14.3	0	0	0	0	0	0	35	0.9	0	0	0	
0	2.6	11	3.5	0	0	0	0	0	0	5.1	5.9	0	0	0	0	
19.8	2	0.6	14.2	0	0	0	4.8	0	0	60.4	0	2.6	0	0	0	
5	8	0	7	0	0	2.3	0	0	0	6.9	0	0	0	0	0	
12.3	1.9	1	0	0	21.5	0	0	0	6.6	3	3	0	0	0	0	
8.1	0.8	2.4	0	0	2.5	1	0	0	0	15.1	60.4	2	0	0	0	
4.8	2.2	13.2	0	0	0	0	0	0	9.5	6	60	0	0	0	0	
5.8	6.5	0.4	0	0	13	0	0		5.1	13.4	76	0	0	0	0	
17	0	0	10.8	0	26.2	0	0		12.5	6.2	79.7	0	0	0	0	
25.1	2.2	1.3	0	10.1	2.2	4.6	5.4		0		65.7	3.5	0	0		
8.3	0	10	6.3	0	3	0	0		0		108	0	0	36.1		
22.9	4.3	2.5	0	4.8	10.5	10			0	3.2	10.5	0	0	41.8		
26.9	0.2	4.6	4	0	0	0	12		0			0	0	30		
0	0	0	19.3	0	0	9.5	0				2.2	0	0	0		
Notes.

Source: Thai Meteorological Department: https://www.tmd.go.th/services/weekly report.php.

Table 6 Daily rainfall data in five Thailand’s regions on August 9, 2019.

Northern	Northeastern	Central	Eastern	Southern	
9.5	0	25.3	20	6.6	8.4	0	67	0	39.6	0	0	27.9	4.1	0.4	114.6	
4.9	10	25.5	14.5	16.9	0.8	2.9	65.4	0	25	0	0	0	9	3.8	0	
0	21.6	24	3	10	20.2	0	21	0	0	0	26.5	3.4	27.3	0.6	0	
4.7	15	8	28	48.2	0	14.3	6.4	7.2	0	0	36.4	0	6.5	0	0	
0	15.5	0	27	6.5	0.5	0	0	3.5	29.7	0.1	0	0.8	3.5	10.8	0	
63.2	14	20	50	4.8	5.3	6	52	0	0	0.3	4.5	37.9	0	5	18.2	
9.6	8.5	0	24	25	16.7	0	45	40.5	3.1	0.5	0	32.4	0	12.2	40.4	
10.7	11.5	0	30	0	45.2	28	41.4	25.8	8.2	31.5	0.5	33.8	0	3.6	0	
13	17.4	0	22	0	0	0	14.3	30.4	3.2	8.2	0.7	15.8	0	0	0	
0	15.6	0	16	3.2	0.6	0	45	0	7.1	0	12.3	0	3.6	8.8	10.8	
0	31.6	33.8	0	44	0	0	27	0	0	0	0.5	0	3	0	0	
0	20.6	33.7	0	0	3.1	27.6	0.2	0	3.2	0	1.9	0	1	0	0	
0	31.1	15.1	0	9.3	33.3	33	30	0	4.2	0	66.4	0	3.7	6.2	35	
0	16.3	18.5	0	0	6	0	0	0	5.7	0	93.6	11.5	15.6	0	0	
2.8	0	44.8	39.7	20	0	0	0	8.3	30	0	68.7	1.7	11.2	3.8	33.5	
11.3	33.1	37.5	9.3	0	13.2	0	0	0	4	0	40	1.2	24	0	57	
0.6	29.2	0	0	4.8	0	0	0	0	0	0	65	21.2	0	0	10.5	
36.1	11.2	47	2.1	0	21	0	0	0	0	0	63.7	0	0	0	0	
0	14.4	20	0	0	0	0	1	36.1	0	0	9.2	30	10.2	0.2	0	
2.6	60	30.8	46.7	0	8.4	15	0	0	0	1.2	0	5.1	0	0	0	
5	42.3	30	10.5	0	0	0	0	12.5	0	0	0	2.5	0	0	30.8	
13.4	9.5	1	0	56.5	0	0	2.5	0	14.7	0.1	11	2.4	0	0.4	10.7	
12.3	34.5	1.2	41	39.2	0.5	0	0	0	0	1	69.6	5	0	0	0	
25.8	36.5	56.3	10.3	0	4.5	25.7	9.5	0	0	3	89.6	1.7	0	0	15.9	
30.2	9.7	0	1.2	6.4	16.2	41.4	0	0	0.5	160		0	0	2.2	0	
16.4	0	6	23.9	5.3	0	41.6	0		1.6	34.3	0	0	0	
6	0	0	22.2	0	3.5	53.8	0		0		0	2.1	0	0.6		
33.1	7.6	5.3	24.1	9.8	20	48.5	0		0		25	0	0	76.6		
16.4	9.6	7.2	38	0	0	78.5	2.1		0		19.5	10.5	7	121.6		
19.8	9.3	24.6	9	9.7	0	12.7		0	0			15.3	10.6	60		
0	0	30	9.2	4.5	1.2	80.9	0		0			0	3.6	0		
Notes.

Source: Thai Meteorological Department: https://www.tmd.go.th/services/weekly_report.php.

Figure 4 Histogram plots of daily rainfall data in five Thailand’s regions on August 5, 2019: (A) Northern (B) Northeastern (C) Central (D) Eastern (E) Southern.

Figure 5 Histogram plots of daily rainfall data in five Thailand’s regions on August 9, 2019: (A) Northern (B) Northeastern (C) Central (D) Eastern (E) Southern.

Figure 6 Normal Q-Q plots of log-positive daily rainfall data in five Thailand’s regions on August 5, 2019: (A) Northern (B) Northeastern (C) Central (D) Eastern (E) Southern.

An empirical application

Daily rainfall data obtained from the Thai Meteorological Department (TMD) were divided into the northern, northeastern, central, and eastern regions, while the southern region was a combination of the data from the southeastern and southwestern shores. Due to the differences in the climate patterns and meteorological conditions in the five regions, we focused was on estimating the daily rainfall data in these regions by treating them as separate sets of observations rather than using the average rainfall for the whole of Thailand by pooling them and treating them as a single population. The daily rainfall amounts were recorded on August 5 and 9, 2019, which is in the middle of the rainy season (mid-May to mid-October) when rice farming is conducted in Thailand. Entries with rainfall of less than 0.1 mm were considered as zero records.

Tables 5–6 contain the daily rainfall records for the five regions, while Figs. 4–5 show histogram plots of rainfall observations, and Figs. 6–7 exhibit normal Q-Q plots of the log-positive rainfall data on August 5 and 9, 2019, respectively. It can be seen that the data for all of the regions contained zero observations. After that, the fitted distribution of the positive observations was checked using the Akaike information criterion (AIC), as reported in Table 7. It can be concluded that the rainfall data in all of the regions on August 5 and 9, 2019 follow a delta-lognormal distribution. All data sets and R code are available in the Supplemental Information. The summary statistics are reported in Table 8. In the approximation of the daily rainfall amounts in the five regions, the estimated common means were 4.4506 and 13.2621 mm/day on August 5 and 9, 2019, respectively. The computed 95% CIs of the common rainfall mean are reported in Table 9. Under the rain criteria issued by the TMD (Department, 2018), it can be interpreted that the daily rainfall in Thailand on August 5, 2019, was light (0.1–10.0 mm), while it was moderate (10.1–35.0 mm) on August 9, 2019. These results confirm the simulation results for k = 5 in the previous section.

Figure 7 Normal Q-Q plots of log-positive daily rainfall data in five Thailand’s regions on August 9, 2019: (A) Northern (B) Northeastern (C) Central (D) Eastern (E) Southern.

Table 7 AIC results of daily rainfall records in five Thailand’s regions.

Regions	AIC	
	Cauchy	Logistic	Lognormal	Normal	T-distribution	
On August 5, 2019	
Northern	373.1958	357.3122	336.8724	353.7757	354.3055	
Northeastern	600.9473	642.1779	543.9619	667.2334	664.6152	
Central	240.0227	266.4162	220.8503	293.9151	283.2302	
Eastern	229.8995	220.2523	202.8394	218.7240	219.1471	
Southern	194.9368	197.5586	178.5587	201.1654	200.1388	
On August 9, 2019	
Northern	389.6257	387.3072	375.7994	391.1802	390.2479	
Northeastern	1123.7491	1080.8694	1052.8953	1080.1467	1079.9365	
Central	178.8516	189.5353	155.0261	190.6855	190.5103	
Eastern	233.5236	227.1725	215.9306	228.0501	227.4559	
Southern	541.0477	569.2615	487.4667	592.2242	588.2377	

Table 8 The summary statistics.

Regions	Estimated parameters	
	ni	μ ˆi	σ ˆi2	δ ˆi	ϑ ˆi	
August 5, 2020	
Northern	62	1.866	1.277	0.210	9.472	
Northeastern	210	1.734	1.578	0.619	4.668	
Central	57	1.085	1.784	0.316	4.741	
Eastern	29	2.366	4.545	0.241	59.391	
Southern	119	1.684	1.730	0.782	2.639	
August 9, 2020	
Northern	62	2.621	0.732	0.226	15.187	
Northeastern	210	2.577	1.502	0.405	16.429	
Central	57	1.190	3.054	0.579	5.542	
Eastern	29	2.860	3.070	0.241	52.813	
Southern	119	2.007	2.051	0.462	10.811	

Table 9 95%CIs of common rainfall mean in five Thailand’s regions.

Methods	95%CIs for ϑ	Lengths	
	Lower	Upper		
On August 5, 2020	
FGCI	2.5545	6.3342	3.7798	
LS	3.2166	5.6846	2.4681	
MOVER	2.7216	9.0296	6.3080	
PB	5.8876	11.4965	5.6089	
HPD-JR	3.5216	7.8533	4.3317	
HPD-NGB	2.4969	6.0904	3.5935	
On August 9, 2020	
FGCI	7.1127	16.8809	9.7682	
LS	10.4880	16.0363	5.5483	
MOVER	7.5814	23.3171	15.7357	
PB	14.5229	23.5821	9.0591	
HPD-JR	12.8404	20.4349	7.5945	
HPD-NGB	7.2928	17.1265	9.8337	

Discussion

It can be seen that for MOVER and PB developed from the studies of Krishnamoorthy & Oral (2015) and Malekzadeh & Kharrati-Kopaei (2019), respectively, the simulation results are similar to both of these studies provided that the zero observations are omitted. CIs for the common mean have been investigated in both normal and lognormal distributions (Fairweather, 1972; Jordan & Krishnamoorthy, 1996; Krishnamoorthy & Mathew, 2003; Lin & Lee, 2005; Tian & Wu, 2007; Krishnamoorthy & Oral, 2015). However, the common mean of delta-lognormal populations is especially of interest because it can be used to fit the data from real-world situations such as investigating medical costs (Zou, Taleban & Huo, 2009; Tierney et al., 2003; Tian, 2005), analyzing airborne contaminants (Owen & DeRouen, 1980; Tian, 2005) and measuring fish abundance (Fletcher, 2008; Wu & Hsieh, 2014). Furthermore, it is possible that some extreme rainfall data also fulfill the assumptions of a delta-lognormal distribution. Note that such natural disasters as floods and landslides have been caused by the extreme rainfall events, as evidenced in many country around the world: Europe (e.g., Northern England, Southern Scotland and Ireland Otto & Oldenborgh, 2017), Asia (e.g., Japan Oldenborgh, 2018) and North America (e.g., Southeast Texas Oldenborgh et al., 2019). Our findings show that some of the methods studied had CPs that were too low or too high for large sample cases, a shortcoming that should be addressed in future work.

Conclusions

The objective of this study was to propose CIs for the common mean of several delta-lognormal distributions using FGCI, LS, MOVER, PB, HPD-JR, and HPD-NGB. The CP and AL as performance measures of the methods were assessed via Monte Carlo simulation. The findings confirm that for small sample case ()k=2 (), FGCI and HPD-NGB are the recommended methods in different situations: FGCI (a small-to-moderate sample size and a large σi2 with a moderate-to-large sample size) and HPD-NGB (small σi2 with a moderate-to-large sample size). For large sample cases (k = 5, 10), MOVER small δi and σi2) and PB (large δi and σi2) performed the best.

Supplemental Information

Supplemental Information 1 Daily rainfall data in five Thailand’s regions on August 5, 2019

Click here for additional data file.

Supplemental Information 2 Daily rainfall data in five Thailand’s regions on August 9, 2019

Click here for additional data file.

Supplemental Information 3 R code for the program for running all outputs.

Click here for additional data file.

The authors are grateful to the academic editor and reviewers for their constructive comments and suggestions which help to improve this manuscript.

Additional Information and Declarations

Competing Interests

Author Contributions

Data Availability

The authors declare there are no competing interests.

Patcharee Maneerat and Sa-Aat Niwitpong conceived and designed the experiments, performed the experiments, analyzed the data, prepared figures and/or tables, authored or reviewed drafts of the paper, and approved the final draft.

The following information was supplied regarding data availability:

All data sets and R code are available in the Supplementary Files.

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
