# Peer review of "Estimating the average daily rainfall in Thailand using confidence intervals for the common mean of several delta-lognormal distributions"

_PeerJ, doi:10.7717/peerj.10758_

## Round 0.1 · original submission · Minor Revisions

In addition to the comments made by the reviewers, in particular by Rev. 1, I would like you to address in plain language the following comment:

Please justify why you consider that the estimate of the average precipitation is the relevant variable to consider. Wouldn't it be more relevant to focus on estimating the extreme daily precipitation? Given that there is observational evidence from many sites around the world there has been an increase in extreme rainfall, and projections for later in the century also indicate increases in extremes, please include some sentences regarding this point, in the introduction and the discussion sections of your manuscript.

Please also make sure that the English text is carefully revised, as Rev. 3 and 4 indicate several errors, which I have also detected.

·

Basic reporting

The text is written clearly and in good English. These authors say what they want to do.

Experimental design

not applicablöe here.

Validity of the findings

I think all findings make sense and are plausible.

Additional comments

Dealing with continuous distributions with a spike at zero is a non-standard problem and it honours the authors to seriously model this situation and the delta-lognormal model is a reasonable approach to cope with the issue.
Here the focus is on estiamting a common mean when data arise from several distributions.
I would like to ask the authors a few questions if I may.
a) In equation (3) the authors define the common mean as a weighted mean where the weights depend on the sample estimate of the respective subpopulation. This is a bit unusal as the population parameter should be defined independently of the sample which is used. A simple solution qwould be define theta identical for all subpopulations.
b) would it then not make sense to define the estiamte as a weighted mean as you intend in (3)?
c) A critical person might argue it would be simpler to pool all the data and treat it as a single population which would be a lot easier. So, I think you need to arge a bit here that your approach is more beneficial. I beleive it is quite similar to meta-analysis.
d) I am not clear with eq. (6) and the arguments leading. The last line in (6) says it is infact unbiased (which I am not clear from the previous line), but why is then there a limitation to obtain an unbiased estimate?

MINOR:
a) I would use a different name for Algorithm 6 as it is not a method that is compared.
b) Presentation of results: it is not clear from tables or grpahs whow the winner is. I would invest a bit more work to illuminate this.
c) application: you do this for a very selectied data set. But if this is done routinely per day or per week, which method should be used taking into acocunt that it must be practicable?
d) I think you need to take this work much more forward. Suppose you have a number of covariates such as time and location how would you take this forward? A way to go could be a two-component model where you model the zeros according to a logistic model and the positive observations with a linear model using a log-transformation on the dependent variable. very much similar to the Lambert model in zero-inflation modelling. But this cleary is for another paper to come.

Reviewer 2 ·

Basic reporting

n/a

Experimental design

n/a

Validity of the findings

n/a

Additional comments

n/a

Reviewer 3 ·

Basic reporting

The Science looks fine but the English needs improvement. More specifically, the Introduction section needs to be edited (re-written).

Experimental design

No comment

Validity of the findings

No comment

Additional comments

Please re-write the introduction section by getting help from a native English speaker.

·

Basic reporting

The article is well clear. Couple of typos were detected. For example: in abstract, there should be space in between 'in' and 'a'. A thorough overview should be given for a flawless article.

Experimental design

'No comment'

Validity of the findings

More discussion on "Simulation study and results" could give the readers a clearer idea.

Additional comments

Overall, this is a very well written article and the problem was well stated. The problem and the conclusions drawn from the article plays a major role in the country like Thailand.
The tables and figures could be labeled properly to give clear idea about each of the cases.
I recommend this article to be accepted with very very minor corrections.

---

## Round 0.2 · Minor Revisions

I received additional comments on your manuscript, which I consider close to acceptance, after these comments are addressed. The revised version will not be sent out to reviewers.
* * *
a) In equation (3) the authors define the common mean as a weighted mean where the weights depend on the sample estimate of the respective subpopulation. This is a bit unusual as the population parameter should be defined independently of the sample which is used. A simple solution would be to define theta identical for all subpopulations.
b) would it then not make sense to define the estimate as a weighted mean as you intend in (3)?
c) A critical person might argue it would be simpler to pool all the data and treat it as a single population which would be a lot easier. So, I think you need to argue a bit here that your approach is more beneficial. I believe it is quite similar to meta-analysis.
d) I am not clear with eq. (6) and the arguments leading. The last line in (6) says it is in fact unbiased (which I am not clear from the previous line), but why is then there a limitation to obtain an unbiased estimate?

MINOR:
a) I would use a different name for Algorithm 6 as it is not a method that is compared.
b) Presentation of results: it is not clear from tables or graphs who the winner is. I would invest a bit more work to illuminate this.
c) application: you do this for a very selected data set. But if this is done routinely per day or per week, which method should be used taking into account that it must be practicable?
* * *
·

Basic reporting

ok

Experimental design

n/a

Validity of the findings

ok

Additional comments

I am happy with the revision.

---

## Round 0.3 · accepted · Accept

I thank the authors for addressing the comments I had made and for clarifying the manuscript where needed.